# A rapid detection tool for VT isolates of *Citrus tristeza virus* by immunocapture-reverse transcriptase loop-mediated isothermal amplification assay

Vijayanandraj Selvaraj[1☯¤]*, Yogita Maheshwari[1☯¤], Subhas Hajeri[2], Raymond Yokomi[1]*

**1** USDA-ARS, San Joaquin Valley Agricultural Sciences Center, Parlier, CA, United States of America,
**2** Citrus Pest Detection Program, Central California Tristeza Eradication Agency, Tulare, CA, United States of America

☯ These authors contributed equally to this work.
¤ Current address: Plant Virus Lab, CSIR-Institute of Himalayan Bioresource Technology, Palampur, Himachal Pradesh, India
* ray.yokomi@ars.usda.gov (RY); vijayanandraj@hotmail.com (VS)

**Data Availability Statement:** All relevant data are within the manuscript and its Supporting Information files. the data underlying the results

## Abstract

Severe strains of *Citrus tristeza virus* (CTV) cause quick decline and stem pitting resulting in significant economic losses in citrus production. A immunocapture reverse-transcriptase loop-mediated amplification (IC-RT-LAMP) assay was developed in this study to detect the severe VT strains that are typically associated with severe CTV symptoms. The sensitivity of RT-LAMP assay was determined by ten-fold serial dilutions of CA-VT-AT39 RNA, in comparison to one-step RT-droplet digital (dd) PCR. RT-LAMP detected up to 0.002 ng RNA with an amplification time of 10:35 (min:sec.), equivalent to 11.3 copies as determined by one step RT-ddPCR. The RT-LAMP assay specifically detected CA-VT-AT39 RNA and did not cross react with other CTV genotypes tested (T36, T30, RB, S1 and T68). To facilitate rapid on-site detection, the RT-LAMP assay was improved by first capturing the CTV virions from citrus crude leaf sap using CTV-IgG (IC-RT-LAMP), thereby eliminating nucleic acid extraction steps. IC-RT-LAMP assay was optimized with two-fold dilutions of CTV-IgG ranging from 1:500 to 1:16,000. The IC-RT-LAMP assay detected the CA-VT-AT39 virions in all dilutions tested. The minimum amplification time was 6:45 (min:sec) with 1:500 and 1:1000 of CTV-IgG dilutions. The limit of detection of IC-RT-LAMP assay with crude leaf sap of CA-VT-AT39 was 1:320 with a maximum amplification time of 9:08 (min:sec). The IC-RT-LAMP assay was validated for VT genotype by comparing to IC-RT-qPCR using the CTV from 40 field tree samples. A 100% agreement was observed between tests, regardless of single or mixed infections of CTV VT with other genotypes. Therefore, the IC-RT-LAMP assay can serve as a useful tool in the management of potentially severe strains of CTV.

presented in this study are available from the USDA, ARS, SJVASC, CDPG, Parlier.

**Funding:** This work was supported by ARS Agreement no. 58-2034-5-026 from Tulare County Pest control District Board, ARS Agreement no. 58-2034-8-003 (5300-185) from Citrus Research Board and Tulare County Pest Control Board, ARS Agreement 58-2034-8-011 (5300) from Citrus Research Board, and ARS Agreement 58-2034-7-018 (Yok 17) California Citrus Nursery Board.

**Competing interests:** The authors have declared that no competing interests exist.

## Introduction

*Citrus tristeza virus* (CTV), a member of the genus *Closterovirus*, family *Closteroviridae*, is a causal agent of the destructive disease named tristeza, one of the most important viral diseases of citrus. CTV is transmitted in California by *Aphis gossypii* in a semi-persistent manner and long-distance movement by infected plant material. CTV induces two economically important disease symptoms: 1) quick-decline (QD) and 2) stem pitting (SP). QD has destroyed millions of citrus trees in Argentina, Brazil, South Africa, California and Spain. The virus exists in infected trees as heterogeneous populations, often with more than one genotype, along with variants and defective RNAs [1–2]. SP induced by CTV is associated with virulent genotypes such as VT, T3 and T68 [3–4]. Sweet orange and grapefruit varieties are the most susceptible scion cultivars, regardless of rootstock. SP severely reduces fruit quality and production [5]. Rootstocks, other than sour orange, offer a solution to the QD problem, but SP is still a concern.

Different techniques have been used to detect CTV, including biological indexing, serology, and nucleic acid-based detection techniques. These techniques are limited to greenhouse or laboratory facilities and require expensive reagents, specialized equipment, and trained personnel to perform the analyses. The development of a rapid and reliable on-site detection method for severe strains of CTV is highly desirable.

LAMP is a sensitive and rapid nucleic acid amplification technology first reported by Notomi *et al*. [6]. It utilizes three sets of forward and reverse oligonucleotide primers specific to six distinct sequences on the target gene. These primers are used to generate amplification products that contain single-stranded loops, thereby allowing primers to bind to these sequences without the need for repeated cycles of thermal denaturation. LAMP assay is performed under isothermal conditions utilizing the *Bacillus stearothermophilus* (*Bst)* DNA polymerase, which has strand-displacement activity [7].

The RT-LAMP was first established for the detection of *Potato virus Y* [8]. The GspSSD LF (OptiGene, UK) polymerases exhibited both reverse transcription and strand displacement polymerase activities, thereby facilitating the development of more rapid and sensitive RT-LAMP systems. The fluorescently labeled amplified product can be quantified using a fluorescence reader, which is light weight, weather-proof, battery operated, GPS enabled, and offers rapid, on-farm end-point detection. Subsequently, RT-LAMP method has been developed for detection of *Potato leaf roll virus* [9], *Pepino mosaic virus* [10], *Tomato torrado virus* [11], and *Southern tomato virus* [12]. Warghane *et al*. [13] showed the sensitivity of RT-LAMP is 100 times more than RT-PCR using the RNA extracted from the CTV infected leaf samples. Immunocapture (IC) RT-LAMP is useful for on-farm detection of pathogens by coating PCR tubes with pathogen specific polyclonal antibodies. IC-RT-LAMP has been developed for *Potato virus Y* [14] and Mirafiori lettuce big-vein virus [15] in lab conditions.

The VT strain of CTV was first described from Israel as a seedling yellows strain [16] with 3' sequences similar to CTV-T36, but with much more variations in the 5' sequences [17]. The "VT-group" is now known to contain CTV isolates that cause stem pitting [3]. However, due to divergent 5' and 3' ends and extensive recombination events, phenotypes of VT strains can be variable, thus, making strain classification based on symptomology alone (mild, stem pitting, severe, etc.) precarious [18]. VT strains in California have been identified by multiple marker method [3,19] or by sequencing [20] and typically cause stem pitting in sweet orange and grapefruit in CTV indexing tests [20]. Ananthakrishnan *et al*. [21] developed a multiplex real-time PCR assay that used VT gene regions of the 5' UTR, ORF 1a, and ORF 2 to detect VT strains. Here, we report further development of a field-deployable IC-RT-LAMP for rapid and efficient detection of VT strains of CTV with high sensitivity.

## Materials and methods

### Primer and probe design

Sixty complete CTV genome sequences available in NCBI GenBank were aligned using the CLUSTALW in MEGA 7 software and used to determine the sequences specific to VT isolates. The CTV sequences were highly variable among the genotypes in the 5' region and were used to design LAMP primers (Table 1) and TaqMan probe (Table 2) specific to VT isolates. PrimerExplorer V5 software was used to design three sets of primers: (i) outer primers: F3/B3, (ii) inner primers: FIP/BIP, and (iii) loop primers: LF/LB (http://primerexplorer.jp/lampv5e/index.html ). TaqMan probes and primers were designed using Primer3 version 4.1.0 software.

### Plant material and RNA isolation

Eight biologically characterized CTV isolates belonging to different genotypes were used to determine the specificity of the LAMP and TaqMan primer and probes. *In planta* CTV isolates CA-VT-AT39, CA-T30-AT4, CA-RB-115, CA-RB-AT35, CA-S1-L, CCTEA 11661 (T36 genotype) [20, 22] were maintained at the USDA-ARS in Parlier, California. Lyophilized citrus leaves infected with B165 and T68 were obtained from Beltsville, Maryland [23]. RNA extraction was done from two hundred mg of leaf petioles using RNeasy Plant Mini Kit (Qiagen) to compare the sensitivity of the RT-LAMP assay with the one step RT-ddPCR assay. The nucleic acid quality and quantity was measured with the Qubit 3.0 fluorometer using the Qubit dsRNA HS assay Kit (Thermo Scientific Inc., Waltham, MA, USA). Field validation of RT-LAMP assay was carried out using the leaf samples collected in May 2018, from four different quadrant branches of 40 citrus trees in seven different orchards within a 2.6 km² area of a Township/Range/Section (U.S. Public Land Survey System) in eastern Fresno and Tulare Counties, California. A full set of genotype-discriminating primers/probes are listed in S1 Table. The exclusivity of the new VT primer set was validated by RT-qPCR with 69 characterized California CTV isolates from an *in planta* CTV collection at the CCTEA and ARS-Parlier and the data is shown in S6 Table.

### RT-LAMP assay

To determine the optimal amplification temperature, the RT-LAMP reactions were carried out in Genie III fluorometer (OptiGene, Horsham, UK) at different temperatures: 59, 60, 61, 62, 63, 64, 65 and 66˚C, with the highest recommended primer concentrations (0.2 µM outer primer, 2 µM inner primer, and 1 µM loop primer). To determine the optimal primer concentration, the RT-LAMP reactions were carried out with varying concentrations of inner primer

**Table 1. Primers used for the RT-LAMP assay for VT strains of *Citrus tristeza virus*.**

| Primer name | Sequence (5'-3') | Primer position (nt)[a] | Primer length (bp) |
|---|---|---|---|
| VT_F3 | CGTACCCTCCGGAAATTACG | 19–38 | 20 |
| VT_B3 | GGTGGTTGTTTCAGTACCGA | 217–236 | 20 |
| VT_FIP (F1c + F2) | AGCGAGCTCCAAGTTTCGACAT–TGCGGGAATTGGTGTAGGT | (108–129)-(48–66) | 22–19 |
| VT_BIP (B1c + B2) | GGTCCTCGACTTTCGCTGTACG–AGTGCAGGACTCCAACGG | (133–154)-(196–213) | 22–18 |
| VT_LF | GGCGTAGTGGGCAATTTGC | 71–89 | 19 |
| VT_LB | TTACGTCATCTCGCGCATCT | 161–180 | 20 |

[a]The primer position numbers are based on CA-VT-AT39 CTV strain.

**Table 2.  New VT-specific primer/probe used for the RT-qPCR and RT-ddPCR assay for VT strains of *Citrus tristeza virus*.**

| Target Gene | Primer/Probe name | Sequence (5'-3') | Primer position | Amplicon size (bp) |
|---|---|---|---|---|
| 5'UTR | VT For | GCTGCGGGAATYGGTGTA | 7–25 | 93 |
| | VT Rev | CGAAAGTCGAGGACYTGAAG | 81–99 | |
| | VT Probe | 6FAM/ CAAATTGCCCACTACGCCCATAC / MGB/NFQ | 51–74 | |

(0.8 μM, 1.2 μM, 1.6 μM and 2 μM) and loop primer (0.5 μM, 0.75 μM and 1 μM) with fixed outer primer concentration in different combinations (Table 3).

The optimized RT-LAMP reaction mixture (25 μl) contained 15 μl of isothermal master mix (ISO 004), 0.25U AMV reverse transcriptase (RT-001) (OptiGen, Horsham, UK), 2.0 μM each of FIP and BIP primers, 1 μM each of LF and LB primers, 0.2 μM each of F3 and B3 primers and 2 μl of template RNA. The amplified product was purified using PCR Clean-up kit (Macherey-Nagel, Düren, Germany) and further confirmed by sequencing. The melt curve starts at 80°C and ends at 98°C with a ramp rate of 0.05°C/sec.

## One step RT-droplet digital PCR (RT-ddPCR) assay

To determine the optimal annealing temperature for VT specific primers in RT-ddPCR assay, thermal gradient ranging from 48, 49, 51, 53.9, 57.2, 60.1, 62.1, and 63°C was performed in the S100 Thermal cycler. The reaction was carried out using the same amount of RNA extracted from leaves of CA-VT-AT39 isolate and primers/probes concentrations (900nM/250nM). The RT-ddPCR was done using one step RT-ddPCR advanced kit for probes (Bio-Rad, CA, USA). The reaction mixture (20 μl) contained 5μl of supermix, 2 μl of reverse transcriptase, 300mM DTT, 900 nM of each forward and reverse primers, 250 nM of TaqMan probe and 2 μl of CTV RNA. The reaction mixture was transferred individually in middle wells of disposable eight channel DG8 cartridge and bottom wells were filled with 70 μl of droplet generation oil. The prepared cartridge was then placed into the QX 200 droplet generator for droplet generation. The prepared droplet emulsions were further loaded in a PCR plate (Eppendorf) using a multi-channel pipette (Rainin, USA). The plate was then heat sealed with pierceable foil using a PX1 PCR plate sealer (Bio-Rad) and PCR amplification was carried out in a S1000 thermal cycler (Bio-Rad). The thermal cycling consisted of reverse transcription at 50°C for 60 min, enzyme

**Table 3.  Optimization of primer concentrations for RT-LAMP assay for VT strains of *Citrus tristeza virus*.**

| Inner primers(FIP/BIP) (μM) | Outer primers(F3/B3) (μM) | Loop primers(LF/LB) (μM) | Time (min:sec) | | | MeanTime (min:sec) | SD |
|---|---|---|---|---|---|---|---|
| | | | R1 | R2 | R3 | | |
| 0.8 | 0.2 | 0.5 | 11:15 | 11:15 | 11:15 | 11:15 | 0.000 |
| 1.2 | | | 9:30 | 9:30 | 9:30 | 9:30 | 0.000 |
| 1.6 | | | 8:30 | 8:30 | 8:30 | 8:30 | 0.000 |
| 2 | | | 8:00 | 8:00 | 8:00 | 8:00 | 0.000 |
| 0.8 | | 0.75 | 10:00 | 10:15 | 10:00 | 10:05 | 0.006 |
| 1.2 | | | 9:00 | 8:45 | 8:45 | 8:50 | 0.006 |
| 1.6 | | | 8:15 | 8:15 | 8:15 | 8:15 | 0.000 |
| 2 | | | 8:00 | 7:45 | 7:45 | 7:50 | 0.006 |
| 0.8 | | **1** | 9:00 | 9:00 | 9:00 | 9:00 | 0.000 |
| 1.2 | | | 8:45 | 9:00 | 8:45 | 8:50 | 0.006 |
| 1.6 | | | 8:00 | 8:00 | 8:00 | 8:00 | 0.000 |
| **2** | | | **7:45** | **7:45** | **7:30** | **7:40** | **0.006** |

activation at 95˚C for 10 min, followed by 40 cycles of 95˚C for 30 s (denaturation) and 56˚C for 1 min (annealing/extension) with a ramp of 2˚C/s and a final 10 min incubation at 98˚C for enzyme deactivation. After thermal cycling, the plate containing the droplets was placed in a QX 200 droplet reader (Bio-Rad, CA, USA) for analysis.

### IC-RT-LAMP assay

To determine the optimum quantity of CTV IgG, the tubes were coated with different concentrations of rabbit CTV IgG (1:500; 1:1000; 1:2000; 1:4000; 1:8000; 1:16000) developed against recombinant CTV virions in carbonate buffer (pH 9.8) and incubated at 37˚ C for 1 hr [24]. After three washes with PBS-T (1X PBS buffer and 0.05% tween-20), the tubes were loaded with leaf extracts, which had been ground in extraction buffer (1X PBS-T and 2% PVP), to trap CTV virions and incubated at 37˚C for 1 hr. The plates were washed as described earlier. The reaction mixture was prepared as described above for RT-LAMP assay by adding water in place of RNA template. The IC-RT-LAMP assay was performed with field samples without extracting RNA (Table 4). Healthy citrus leaf extract was used a negative control.

### One step IC-RT-qPCR assay

The CTV infected field samples were genotyped for single or mixed infection by RT-qPCR using CTV strain discriminating primers and TaqMan hydrolysis probes as listed in S1 Table. One step RT-qPCR assay was performed in the CFX96 Real-Time System (Bio-Rad) using iTaq™ Universal Probes One-Step Kit (Bio-Rad). The reaction mixture contained 2x iTaq universal probe mix, 0.5 µl of iScript advanced reverse transcriptase, 300 nM forward and reverse primers, 150 nM probe, 2 µl of CTV RNA and final volume made up to 20 µl with double distilled water. The thermal cycling conditions consisted of reverse transcription for 10 min at 50˚C, enzyme activation at 95˚C for 5 min, then 40 cycles of denaturation at 95˚C for 10 s, and annealing at 56˚C for 30 s. Each run included CTV positive and RNA from healthy leaves were used as negative controls.

## Results

### Optimization of RT-LAMP assay

The optimal amplification temperature 65˚C was selected based on gradient RT-LAMP assay having the lowest amplification time of 8:15 (min:sec) (Fig 1A, S2 Table). The selected

**Table 4. Comparison of genotyping of *Citrus tristeza virus*-positive field trees tested in central California by TaqMan hydrolysis probe versus IC-RT-LAMP.**

| Location | County | TRS[a] | No. trees tested | MCA13 positive | No. trees per genotype group[b] | | | No. VT positive by IC-RT-LAMP/site |
|---|---|---|---|---|---|---|---|---|
| | | | | | VT alone | VT/T30 mixture | T30 alone | |
| Orange Cove | Fresno | 15/24/22 | 5 | 0 | 0 | 0 | 5 | 0 |
| Seville | Tulare | 17/25/14 | 10 | 10 | 10 | 0 | 0 | 10 |
| Orosi | Tulare | 16/25/15 | 5 | 5 | 5 | 0 | 0 | 5 |
| Lindcove | Tulare | 18/27/18 | 5 | 5 | 0 | 5 | 0 | 5 |
| Lindsey | Tulare | 20/27/18 | 5 | 0 | 0 | 0 | 5 | 0 |
| Strathmore | Tulare | 20/27/27 | 5 | 0 | 0 | 0 | 5 | 0 |
| Porterville | Tulare | 22/27/09 | 5 | 0 | 0 | 0 | 5 | 0 |
| Totals | | | 40 | 20 | 15 | 5 | 20 | 20 |

[a]Township/Range/Section (T/R/S) http://www.jsu.edu/dept/geography/mhill/phygeogone/trprac.html
[b]Field trees were infected by single or mixture to CTV genotypes.

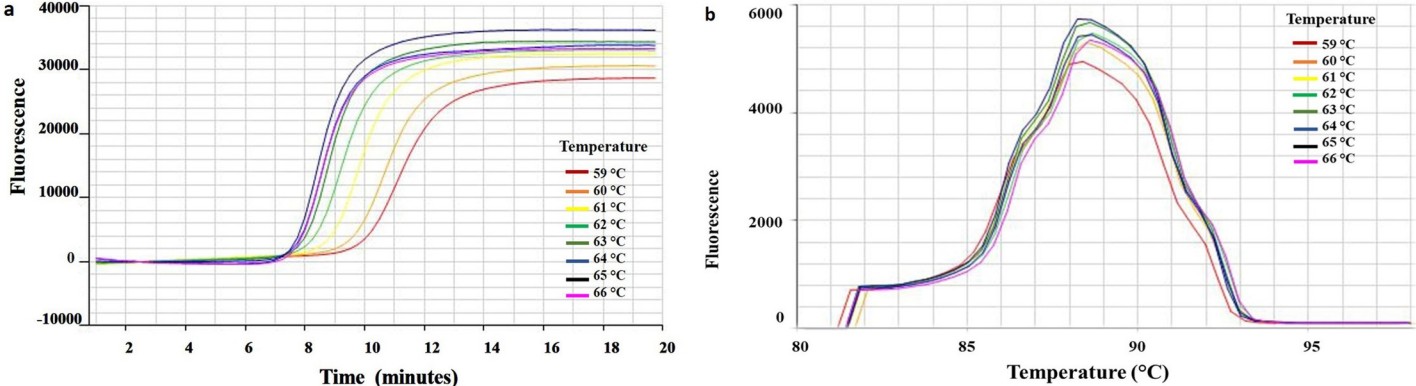

**Fig 1. Amplification curves of thermal gradient RT-LAMP for optimizing annealing temperature for *Citrus tristeza virus* (CTV) VT isolate specific LAMP primers.** (a) RT-LAMP reactions with an annealing temperature gradient ranged from 59˚C to 66˚C was set using the Genie III. CA-VT-AT39 RNA detection times were 11:00, 10:35, 9:40, 9:00, 8:35, 8:20, 8:15, and 8:25 (min:sec) at temperatures 59˚C, 60˚C, 61˚C, 62˚C, 63˚C, 64˚C, 65˚C, 66˚C, and 67˚C, respectively. (b) Melting curve of thermal gradient RT-LAMP primers of *Citrus tristeza virus* CA-VT-AT39 isolate after amplification for 20 min and detected on the FAM channel using 1˚C steps, and a hold of 30 sec at each step from 59 to 96˚C. The viral RNA had a melting temperature (Tm) of 88.4±0.2˚C indicating similar sequences and amplicon size.

temperature had the highest fluorescence of 8583.02 at a melting point of 88.6˚C for the VT specific RT-LAMP primers (Fig 1B).

The inner primers having a concentration of 2 µM and fixed 0.2 µM of outer primers showed the lowest amplification times of 8:00, 7:50, and 7:40 (min:sec)with 0.5, 0.75, and 1 µM of loop primers, respectively. The primer combination of 0.2 µM outer primer, 2 µM inner primer, and 1 µM of loop primers was selected for further experiments to validate the RT-LAMP assay (Table 3).

## RNA dilution

The CA-VT-AT39 isolate was ten-fold serially diluted and subjected to detection by RT-LAMP assay (Fig 2A) and compared with one-step RT-ddPCR (Fig 2B) to measure the absolute copy numbers of viral RNA. The RT-LAMP was able to detect the viral RNA up to five-fold

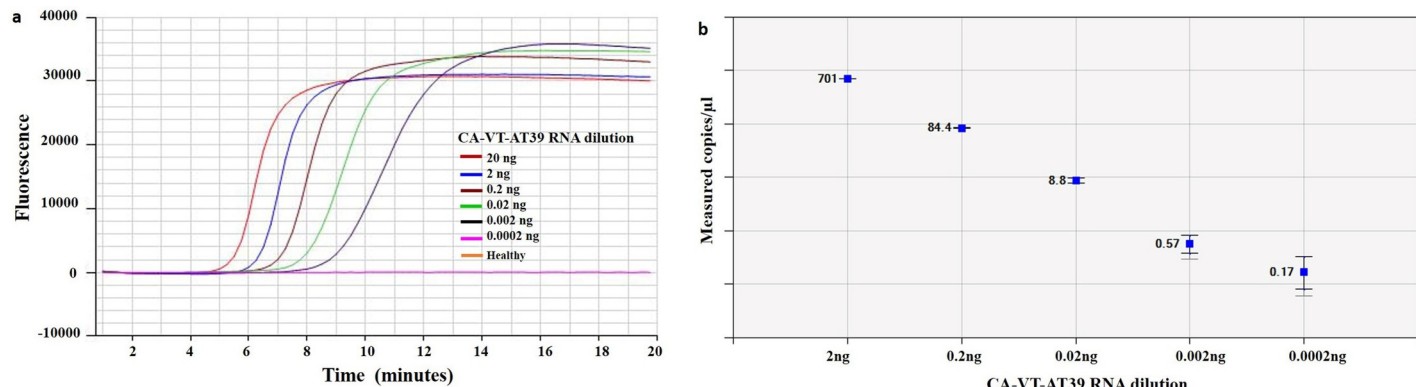

**Fig 2. Amplification curve and Linear regression of tenfold serially diluted CA-VT-AT39 infected citrus leaf RNA in RT-LAMP assay and one step RT-ddPCR for a VT strain of *Citrus tristeza virus*.** (a) Amplification curve of healthy citrus tissue RNA and ten-fold serially diluted ranging from 20 ng to 0.0002 ng of CA-VT-AT39 infected citrus leaf tissue RNA in RT-LAMP assay. CA-VT-AT39 RNA detection times were 6:25, 7:10, 8:10, 9:10, 10:35 (min:sec) for 20 ng, 2 ng, 0.2 ng, 0.02 ng, 0.002 ng of RNA respectively. (b) The Pearson correlation coefficient of CA-VT-AT39 RNA the regression curves (y = 0.967x—182.68) is 0.9998. The inner error bars indicate the Poisson 95% confidence interval (CI) and the outer error bars show the total 95% CI of replicates.

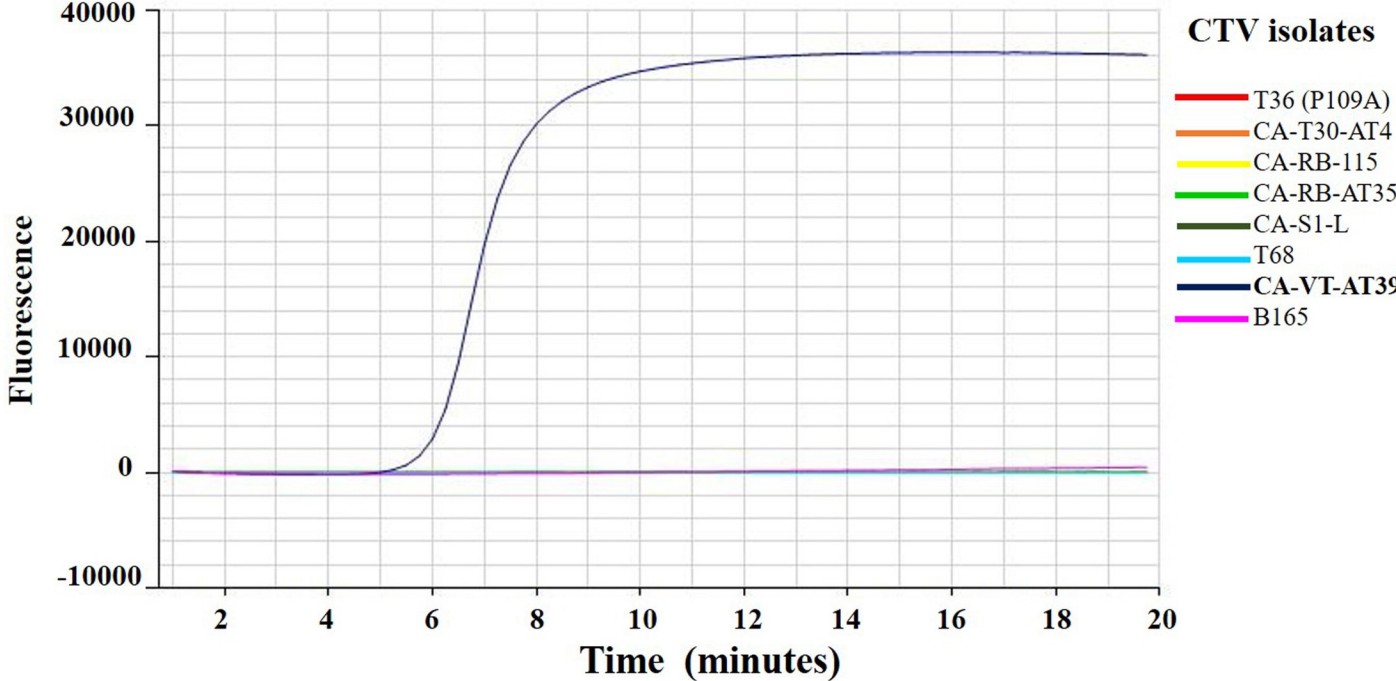

**Fig 3. Amplification curves of eight *Citrus tristeza virus* (CTV) isolates RNA belonging to different genotypes using RT-LAMP assay.** Amplification curves of CTV isolates T36 (P109A), CA-T30-AT4, CA-RB-115, CA-RB-AT35, CA-S1-L, T68, CA-VT-AT39, and B165 using RNA extracted from infected citrus leaves in the RT-LAMP assay. CA-VT-AT39 RNA detection time was 5:58 (min:sec) and no amplification was observed with other strains.

dilutions ranging from 20 ng to 0.002 ng. The detection limit of the RT-LAMP was 11.4 copies as determined by one step RT-ddPCR, with an amplification time of 10:35 (min:sec) (S3 Table).

## Specificity of VT-LAMP primers

The VT primers specifically detected CA-VT-AT39 RNA in the RT-LAMP assay and did not have any cross reaction with the other CTV genotype strains (Fig 3).

## Optimization of CTV IgG for IC-RT-LAMP assay

IC-RT-LAMP was developed for specific detection of VT viral RNA in the citrus crude leaf extracts (Fig 4). The CA-VT-AT39 infected citrus leaf extracts were ground in carbonate buffer (pH 9.8) at a ratio of 1:10. The crude leaf extracts were subjected to IC-RT-LAMP with different concentrations of CTV-IgG to determine the optimum concentrations of CTV-IgG. The viral RNA was detected at all the concentrations of antibody ranging from 1:500 to 1:16000. The antibody concentrations 1:500 and 1:1000 detected the viral RNA with a minimum amplification time of 6:27 (min:sec). Whereas the antibody concentrations 1:2000, 1:4000, 1:8000 and 1:16,000 detected the viral RNA with an amplification times of 7:03, 8:09, 8:27 and 9:00 (min:sec), respectively (S4 Table). The antibody dilution 1:1000 was, thus, selected and used to screen for VT-CTV in field samples.

## Optimization of sample dilution for IC-RT-LAMP assay

To determine the optimal ratio of crude leaf extracts for IC-RT-LAMP assay, extracts from CA-VT-AT39 were diluted to different ratios and subjected to IC-RT-LAMP assay with 1:1000

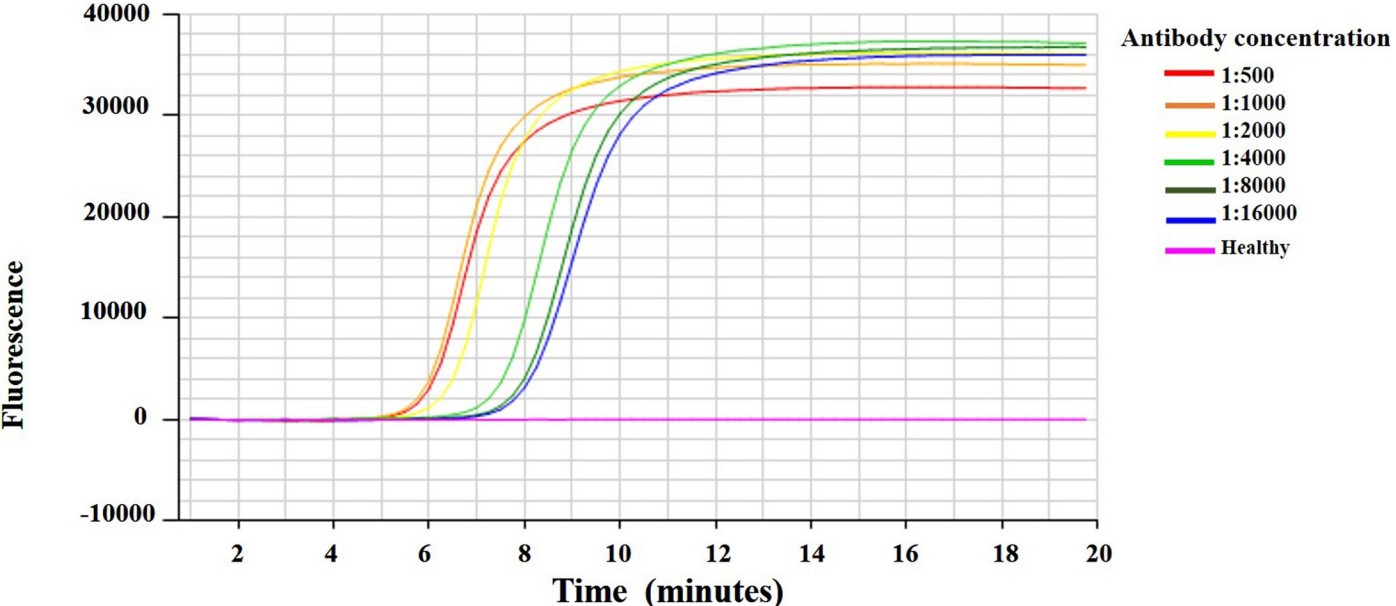

**Fig 4. Optimization of *Citrus tristeza virus* (CTV) IgG against the CA-VT-AT39 infected citrus leaf crude extracts by Immunocapture RT-LAMP assay.** Amplification curve of healthy and CA-VT-AT39 infected citrus leaf tissue extract using two-fold serially diluted (1:500 to 1:16,000) CTV specific antibody in IC-RT-LAMP assay. VT detection times for the CA-VT-AT39 extract were 6:45, 6:45, 7:05, 8:15, 8:45, and 9:00 (min:sec) at 1:500, 1:1000, 1:2000, 1:4000, 1:8000 and 1:16000, respectively.

CTV IgG (Fig 5). The viral RNA was detected at all the dilutions ranging from 1:10 to 1:320. At a dilution of 1:10, the minimum amplification time was 5:12 (min:sec), whereas at dilutions of 1:20, 1:40, 1:80, 1:160, and 1:320, the amplification times were 6:00, 6:36, 7:27, 8:27, and 9:48 (min:sec), respectively (S5 Table). Hence, the 1:10 dilution was selected for screening of VT-CTV infected in field samples.

## Validation of the IC-RT-LAMP

CTV genotypes of field isolates were determined by TaqMan hydrolysis probes in one step IC-RT-qPCR (Table 4) for forty CTV infected field samples from seven different sites in central California. VT hydrolysis probe detected VT isolates in fifteen samples as single infection and 5 samples as mixture of VT and T30 genotypes. IC-RT-LAMP was also carried out for the same samples using VT specific LAMP primers. IC-RT-LAMP assay showed positive amplification in all twenty VT samples that were positive for VT probe in IC-RT-qPCR. The IC-RT-LAMP did not show any cross reactivity with the VT negative samples in IC-RT-qPCR. Hence, the IC-RT-LAMP assay successfully detected VT isolates. Additional validation of the new VT primer/probe was conducted on 43 CTV isolates from California by RT-qPCR. The new VT primer/probe was 100% in concordance with 20 VT and VT mixtures as positive and with 23 non-VT isolates as negative. Full data are shown in S6 Table.

## Discussion

CTV severe strains cause quick decline and stem pitting and these diseases cause economic losses to citrus worldwide. Severe CTV strains are typically associated with VT, T3, and T68 CTV genotypes. These genotypes cause severe to moderate stem pitting symptoms in most of the citrus varieties, irrespective of resistant or tolerant rootstocks. VT is the only severe

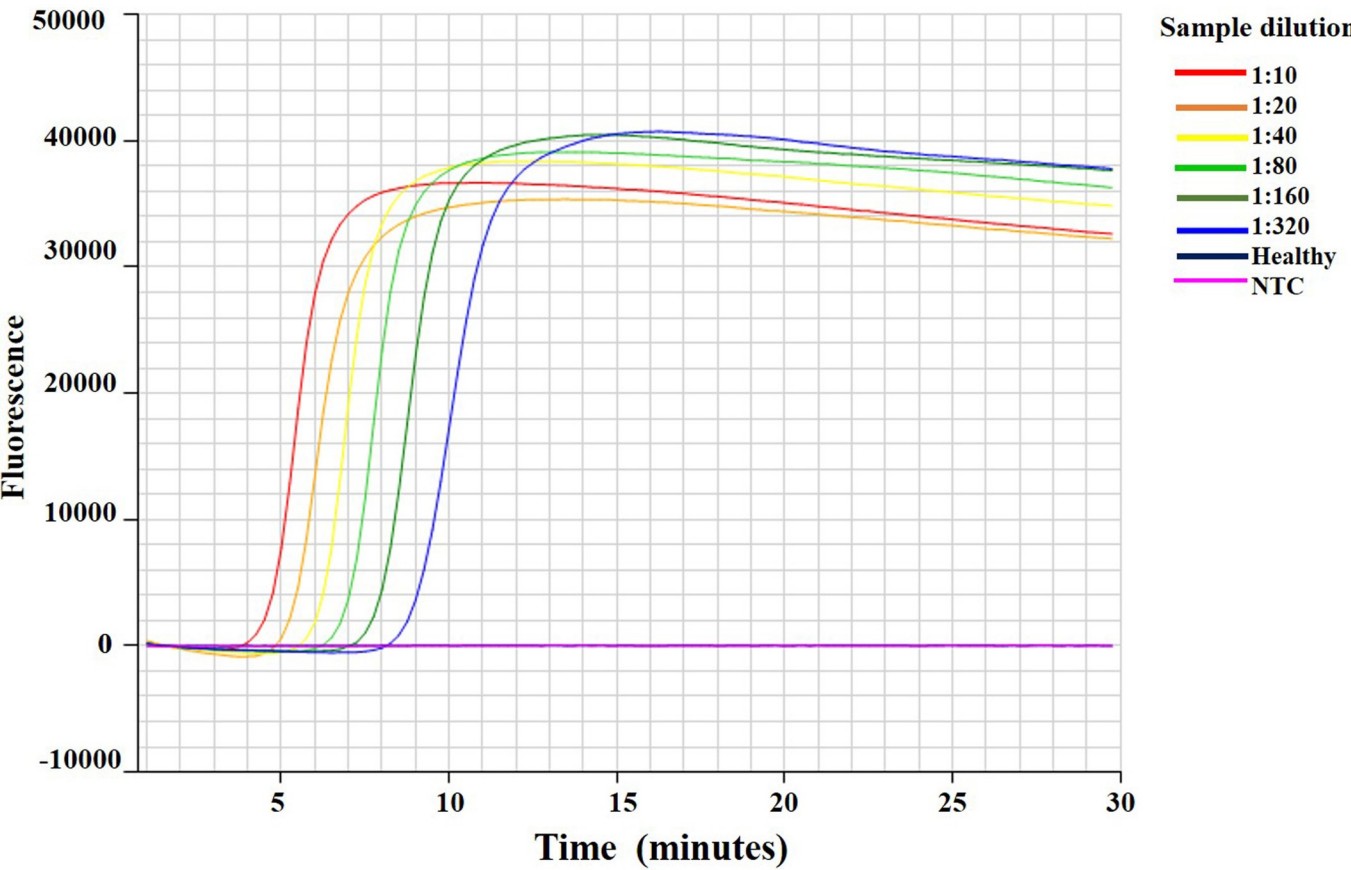

**Fig 5. Optimization of *Citrus tristeza virus* infected leaf extract dilution for IC-RT-LAMP assay.** Amplification curves IC-RT-LAMP assay of extracts from healthy citrus tissue and two-fold serially diluted (1:10 to 1:320) leaf extract from CA-VT-AT39 infected citrus shows VT detection times for VT were 5:12, 6:00, 6:38, 7:27, 8:27, and 9:49 (min:sec) at different dilution ratios of 1:10, 1:20, 1:40, 1:80, 1:160, and 1:320, respectively.

genotype that has been reported in California. CTV is a quarantine pathogen in California and is being monitored in the Citrus Pest Control Districts in central California by the Central California Tristeza Eradication Agency (CCTEA). The CCTEA surveys and screens for severe CTV strains by ELISA using the MCA13 monoclonal antibody [25]. MCA13 monoclonal antibody reacts to all severe CTV genotypes tested including those with VT genotype [26]. In California, MCA13 positive trees are considered to harbor potential severe strains and, as such, are subject to regulatory actions such as tree removal. CTV genotypes S1 and RB also react to MCA13 [20, 22], but biocharacterization has shown these California isolates are mild and do not induce stem pitting or seedling yellows reaction in virus indexing tests. However, implementation of this regulatory program requires these mild strains to be removed. Currently, RT-qPCR using CTV genotype specific probes are available to detect VT and other CTV genotypes [4, 20, 23, 27–29]. Another method combining sequential enzyme immunoassays and capillary electrophoreses-single strand conformation polymorphisms can be used to characterize CTV isolates [30]. However, these techniques are expensive, complex, time-consuming and results typically require days, weeks, or more to reach the grower. In this study, we have developed a one-step IC-RT-LAMP assay for on-site detection of the VT genotype.

The RT-LAMP assay has been shown to be more sensitive than RT-PCR for detection of plant viruses in several cases [31–32]. RT-LAMP assay doesn't require any specialized equipment and the Genie III fluorometer allowed us to monitor the real time fluorescence for quick

and routine detection of CTV. Initially, we developed an *in vitro* RT-LAMP assay with the Genie III fluorometer. The assay was standardized using the RNA extracted from CA-V-T-AT39 infected citrus leaves. The detection limit of RT-LAMP assay was quantified using one-step RT-ddPCR. The RT-LAMP detected the CA-VT-AT39 RNA molecules up to 11.4 copies, whereas RT-ddPCR detected 3.4 copies of CA-VT-AT39 RNA in an aliquot from the same sample. The RT-LAMP specifically detected the CA-VT-AT39 RNA and didn't react with other CTV genotypes reported in California (T30, T36, S1, RB) and elsewhere (T68).

IC-RT-LAMP assay was adapted for on-site detection of CTV VT genotype using the CTV IgG [24]. This eliminates the RNA extraction step and reduces the turnaround time and cost involved. The minimum quantity of CTV IgG and CTV infected crude extract dilution required for IC-RT-LAMP assay was optimized using two-fold serial dilutions of CTV-IgG and CA-VT-AT39 infected leaves, respectively. The IC-RT-LAMP assay was validated using CTV infected positive samples obtained from seven different locations in central California. CTV indexing using genotype discriminating TaqMan hydrolysis probe by RT-qPCR showed that the IC-RT-LAMP assay specifically reacted to the VT genotype. IC-RT-LAMP assay did not show any cross reaction with crude leaf extracts of healthy citrus and other CTV genotypes in mixed infected plants. In addition to IC-RT-LAMP assay, the TaqMan hydrolysis probe was shown to be specific to VT genotype in RT-qPCR and one step RT-ddPCR.

The IC-RT-LAMP assay developed in this study provides real time on-site detection of VT genotype. Although phenotypes of VT isolates can vary significantly in severity [18], rapid screening for VT strain isolates is a tool to assess priorities for CTV management. The IC-R-T-LAMP assay is simple, specific, and more economic than RT-qPCR. The assay can be performed in the field by unskilled labor within an hour and a half using the Genie fluorometer. Thus, the LAMP assay can be used for timely removal of potential severe stains of CTV.

## Supporting information

**S1 Table. RT-qPCR primers and probes used in a matrix to genotype *Citrus tristeza virus* isolates in this study.**
(DOCX)

**S2 Table. Temperature gradient of reverse transcriptase LAMP assay for detection of CA-VT-AT39 strain of *Citrus tristeza virus*.**
(DOCX)

**S3 Table. Detection of CA-VT-AT39 RNA by RT-ddPCR and RT-LAMP assays.**
(DOCX)

**S4 Table. Optimization of *Citrus tristeza virus* polyclonal antibody concentration for Immuno-capture RT-LAMP assay for VT-CTV detection.**
(DOCX)

**S5 Table. Dilution of citrus leaf crude extract by IC-RT-LAMP assay for detection of VT strains of *Citrus tristeza virus*.**
(DOCX)

**S6 Table. Validation of new primer and hydrolysis probe for VT genotype strains of 43 isolates of *Citrus tristeza virus* (CTV) isolates by Reverse Transcription-quantitative Polymerase Chain Reaction Assay (RT-qPCR) in comparison with a matrix of other genotype-specific primer/probes.** [a]TRS: Township/Range/Section is basic unit in the US Public Land Survey System and it is a square piece of land one mile by one mile containing 640 acres. http://www.jsu.edu/dept/geography/mhill/phygeogone/trprac.html. [b]CTV strain

differentiation was performed with four-sets of duplex RT-qPCR tests with specific primer pair and hydrolysis probes as follows: 1) CP-CY5 and VT (new)-VIC; 2) RB/T36-VIC and T36-FAM; 3) S1-FAM and RB/S1-TET; 4) VT3-FAM and T30-VIC (data in bold). [c]New VT primer/probe validated by side-by side comparison with the VT3 primer/probe (data in bold). [d]NA = No Reaction. [e]This T/R/S locale has no section designation.
(DOCX)

## Acknowledgments

We thank Robert DeBorde and Casey Crockett of the United States Department of Agriculture-Agricultural Research Service, San Joaquin Valley Agricultural Sciences Center, Parlier, CA for technical assistance and Rachel Rattner of the United States Department of Agriculture-Agricultural Research Service, San Joaquin Valley Agricultural Sciences Center, Parlier, CA for assistance in editing of the manuscript. Access to the field plot was granted under a Confidentiality Agreement for Entry into California Citrus Orchards between the grower and the U.S. Department of Agriculture, Agricultural Research Service (USDA, ARS). Mention of trade names or commercial products in this publication is solely for providing specific information and does not imply recommendation or endorsement by the USDA. USDA is an equal opportunity provider and employer.

## Author Contributions

**Conceptualization:** Vijayanandraj Selvaraj, Yogita Maheshwari, Raymond Yokomi.

**Data curation:** Vijayanandraj Selvaraj, Yogita Maheshwari.

**Formal analysis:** Vijayanandraj Selvaraj, Yogita Maheshwari.

**Funding acquisition:** Raymond Yokomi.

**Investigation:** Vijayanandraj Selvaraj, Yogita Maheshwari, Subhas Hajeri, Raymond Yokomi.

**Methodology:** Vijayanandraj Selvaraj, Yogita Maheshwari, Subhas Hajeri, Raymond Yokomi.

**Project administration:** Subhas Hajeri, Raymond Yokomi.

**Resources:** Raymond Yokomi.

**Software:** Vijayanandraj Selvaraj, Yogita Maheshwari, Raymond Yokomi.

**Supervision:** Subhas Hajeri, Raymond Yokomi.

**Validation:** Vijayanandraj Selvaraj, Yogita Maheshwari, Subhas Hajeri, Raymond Yokomi.

**Visualization:** Vijayanandraj Selvaraj, Yogita Maheshwari.

**Writing – original draft:** Vijayanandraj Selvaraj, Yogita Maheshwari, Raymond Yokomi.

**Writing – review & editing:** Vijayanandraj Selvaraj, Yogita Maheshwari, Subhas Hajeri, Raymond Yokomi.

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
