## [Editor Report · Decision Letter 0]

18 Jun 2019

PONE-D-19-14855

A rapid detection tool for VT isolates of Citrus tristeza virus by immunocapture-reverse transcriptase loop-mediated isothermal amplification assay

PLOS ONE

Dear Research Plant Pathologist Yokomi,

Thank you for submitting your manuscript to PLOS ONE. After careful consideration, we feel that it has merit but does not fully meet PLOS ONE’s publication criteria as it currently stands. Therefore, we invite you to submit a revised version of the manuscript that addresses the points raised during the review process.

Please see the Academic Editor Comments below.

We would appreciate receiving your revised manuscript by Aug 02 2019 11:59PM. To enhance the reproducibility of your results, we recommend that if applicable you deposit your laboratory protocols in protocols.io, where a protocol can be assigned its own identifier (DOI) such that it can be cited independently in the future. For instructions see: http://journals.plos.org/plosone/s/submission-guidelines#loc-laboratory-protocols

We look forward to receiving your revised manuscript.

Kind regards,

Ulrich Melcher

Academic Editor

PLOS ONE

**Journal Requirements:**

2.

We noticed you have some minor occurrence of overlapping text with the following previous publication(s), which needs to be addressed:

https://apsjournals.apsnet.org/doi/10.1094/PHYTO-100-4-0319

https://www.nature.com/articles/s41598-017-13881-4

In your revision ensure you cite all your sources (including your own works), and quote or rephrase any duplicated text outside the methods section. Further consideration is dependent on these concerns being addressed."

3. In your Methods section, please provide additional location information, including geographic coordinates for the data set if available.

4.We note that you have stated that you will provide repository information for your data at acceptance. Should your manuscript be accepted for publication, we will hold it until you provide the relevant accession numbers or DOIs necessary to access your data. If you wish to make changes to your Data Availability statement, please describe these changes in your cover letter and we will update your Data Availability statement to reflect the information you provide.

**Additional Editor Comments (if provided):**

Please could you address the following comments before your manuscript is sent out for external review:

Generalities : The title mentions the VT strain, yet there are absolutely no references to VT genomic uniqueness especially in its 5'end as demonstrated already in 1996 following its complete sequencing (Mawassi et al 1996 ) and comparison with T36  (Karasev et al, 1995). 

Similarly nothing is mentioned on VT :  unique biological characteristics, subdivision into  different  isolates reported which cause r both quick decline QD  and stem pitting (SP), while  others cause  SP  of grapefruits with minimal or no effects on orange trees grafted on sour oranges. 

On the specifics, the authors did mention previous reports  on CTV genotyping by Northern Hybridization  and sequential analyses of enzyme immunoassays and capillary electrophoresis-single-strand conformation polymorphisms. J Virol Methods. 181:139-47. 

Furthermore in the above JVM paper  tests were conducted on RNA recovered from positively  reacting ELISA wells , thus directing genotyping only for the positively reacting citrus trees

A previous  report on RT-LAMP assay for sensitive detection of CTV  by Warghane in J Virol Methods. 2017 Dec;250:6-10. doi: 10.1016/j.jviromet.2017.09.018. Epub 2017 Sep 21 is also missing. 
---

## [Author Response · Author response to Decision Letter 0]

11 Jul 2019

1. Address format corrected

2. overlapping text. I believe this was corrected by adding the appropriate reference

3. Rather than geographic coordinates, Township/Range/Section was added for each field collection.

4. We don't need repository information. The relevant data is included in our submission.

5. VT description expanded and references added.

6. VT phenotype variation was also added

7. CTV genotyping. This study was not to differentiate CTV genotypes, rather, to screen CTV isolates efficiently for potential severe phenotype which has a VT genotype. 

8. Immunocapture. Yes, the CCTEA screens CTV samples from infected trees with MCA13 monoclonal antibody and extracts from MCA13-positive samples are sent to my ARS lab for molecular genotyping by RT-qPCR with strain discriminating primers/probes and selected isolates are biocharacterized in a virus index using graft inoculation to a citrus host range. 

9. Warghane et al 2017 was mentioned in the introduction of the original submission and is still there.

---

## [Editor Report · Decision Letter 1]

17 Jul 2019

PONE-D-19-14855R1

A rapid detection tool for VT isolates of Citrus tristeza virus by immunocapture-reverse transcriptase loop-mediated isothermal amplification assay

PLOS ONE

Dear Research Plant Pathologist Yokomi,

Thank you for submitting your manuscript to PLOS ONE. After careful consideration, we feel that it has merit but does not fully meet PLOS ONE’s publication criteria as it currently stands. Therefore, we invite you to submit a revised version of the manuscript that addresses the points raised during the review process.

We would appreciate receiving your revised manuscript by Aug 31 2019 11:59PM. To enhance the reproducibility of your results, we recommend that if applicable you deposit your laboratory protocols in protocols.io, where a protocol can be assigned its own identifier (DOI) such that it can be cited independently in the future. For instructions see: http://journals.plos.org/plosone/s/submission-guidelines#loc-laboratory-protocols

We look forward to receiving your revised manuscript.

Kind regards,

Ulrich Melcher

Academic Editor

PLOS ONE

Additional Editor Comments (if provided):

The manuscript has too many language errors to be accepted as is. Here is a list of necessary changes. A proof reader should be asked to deal with other possible problems.

l. . 116 "leaf samples"l.121 number disagreement dara are

l.121 Table S5

l. 124-5 temperatures

l. 164 were used as negative

L. 238 "extracts was grounded ???? were ground

l. 245 used in further.

l. 252 times were

l.256 at different ratios

New comments on second review:

The language usage still requires considerable work before the manuscript is acceptable. Research is fine.

l. 92 sequences...were used to

l. 113 articles "the" or plural needed

L. 129 change "at" to "in"

l. 224 ia not a sentence

l.226 number disagreement Primeres...doesn't

also in line 259

l.238 "was ground"

l.273 "data are shown"

l. 287

l. 298-99

l.326 delete "an"

---

## [Author Response · Author response to Decision Letter 1]

24 Jul 2019

All editor call-outs in Additional Editor Comments and New Comments on Second Review corrected and additional grammar and missing information added. Time units for amplification time was standardized and express as min:sec.

---

## [Editor Report · Decision Letter 2]

23 Aug 2019

A rapid detection tool for VT isolates of Citrus tristeza virus by immunocapture-reverse transcriptase loop-mediated isothermal amplification assay

PONE-D-19-14855R2

Dear Dr. Yokomi,

We are pleased to inform you that your manuscript has been judged scientifically suitable for publication and will be formally accepted for publication once it complies with all outstanding technical requirements.

With kind regards,

Ulrich Melcher

Academic Editor

PLOS ONE

Additional Editor Comments (optional):

I appreciate all the work on language improvement
---

## [Editor Report · Acceptance letter]

27 Aug 2019

PONE-D-19-14855R2 

A rapid detection tool for VT isolates of Citrus tristeza virus by immunocapture-reverse transcriptase loop-mediated isothermal amplification assay 

Dear Dr. Yokomi:

I am pleased to inform you that your manuscript has been deemed suitable for publication in PLOS ONE. Congratulations! Your manuscript is now with our production department. 

With kind regards,

on behalf of

Dr. Ulrich Melcher 

Academic Editor

PLOS ONE